# Incorporation of Dual-Stimuli Responsive Microgels in Nanofibrous Membranes for Cancer Treatment by Magnetic Hyperthermia

**DOI:** 10.3390/gels7010028

**Published:** 2021-03-05

**Authors:** Adriana Gonçalves, Filipe V. Almeida, João Paulo Borges, Paula I. P. Soares

**Affiliations:** CENIMAT/i3N, Materials Science Department, NOVA School of Science and Technology, Campus da Caparica, NOVA University of Lisbon, 2829-516 Caparica, Portugal; aml.goncalves@campus.fct.unl.pt (A.G.); fvm.almeida@fct.unl.pt (F.V.A.); jpb@fct.unl.pt (J.P.B.)

**Keywords:** colloidal electrospinning, magnetic hyperthermia, magnetic nanoparticles, microgel, poly (*N*-isopropyl acrylamide), poly (vinyl pyrrolidone)

## Abstract

The delivery of multiple anti-cancer agents holds great promise for better treatments. The present work focuses on developing multifunctional materials for simultaneous and local combinatory treatment: Chemotherapy and hyperthermia. We first produced hybrid microgels (MG), synthesized by surfactant-free emulsion polymerization, consisting of Poly (*N*-isopropyl acrylamide) (PNIPAAm), chitosan (40 wt.%), and iron oxide nanoparticles (NPs) (5 wt.%) as the inorganic component. PNIPAAm MGs with a hydrodynamic diameter of about 1 μm (in their swollen state) were successfully synthesized. With the incorporation of chitosan and NPs in PNIPAAm MG, a decrease in MG diameter and swelling capacity was observed, without affecting their thermosensitivity. We then sought to produce biocompatible and mechanically robust membranes containing these dual-responsive MG. To achieve this, MG were incorporated in poly (vinyl pyrrolidone) (PVP) fibers through colloidal electrospinning. The presence of NPs in MG decreases the membrane swelling ratio from 10 to values between 6 and 7, and increases the material stiffness, raising its Young modulus from 20 to 35 MPa. Furthermore, magnetic hyperthermia assay shows that PVP-MG-NP composites perform better than any other formulation, with a temperature variation of about 1 °C. The present work demonstrates the potential of using multifunctional colloidal membranes for magnetic hyperthermia and may in the future be used as an alternative treatment for cancer.

## 1. Introduction

Cancer is a severe disease with increasing incidence and related deaths, being considered the second cause of death worldwide. According to cancer statistics, this disease was responsible for an estimated 9.6 million deaths in 2018 [1]. The growth and aging of the population as well as the increase prevalence of established risk factors such as smoking, overweight, physical inactivity, and changing reproductive patterns associated with urbanization and economic development are some of the factors responsible for the increase of new cancer cases [2]. Although the available techniques for early detection of cancer have significantly improved in the last decades, late-stage presentation and diagnostics still occur more frequently than expected. Additionally, current cancer treatments are focused on surgery (about 50%), chemotherapy, and radiotherapy, which present a low degree of personalization to the cancer type, leading to their therapeutic inefficiency and severe side effects [1,3]. Over the past decade, advanced therapies have been developed, namely targeted therapies [4] and immune checkpoint therapies [5], which target cancer cells and their microenvironment more specifically, showing superior results to conventional chemotherapy. Despite the optimistic results, there are often problems with resistance and levels of toxicity (side effects) [6]. This can be attributed, at least in part, to the poor local delivery of therapeutic agents, which makes the development of multifunctional systems that deliver medicines locally to the tumor a promising area of research in cancer. The ability to combine different systems and their various properties widens the field of the device’s potential applications, allowing a more personalized treatment.

Thermosensitive polymers are part of a group of “smart” materials sensitive to the application of an external stimulus. These polymers can respond to a temperature variation, changing their conformational structure when in the presence of a suitable solvent. One of the most studied thermosensitive polymers is Poly (*N*-isopropyl acrylamide) (PNIPAAm) since it has a negative temperature response, with a Lower Critical Solution Temperature (LCST) near human body temperature (30–32 °C) [7,8]. This means that bellow that temperature the microgels are in their swollen state and collapse when the temperature rises above LCST. The thermoresponsive behavior showed by PNIPAAm emerges from the interactions between its hydrophilic and hydrophobic groups—acrylamide and isopropyl groups, respectively. Below the LCST, the hydrophilic groups of the polymer establish intermolecular hydrogen bonds with water, leading the microgels to be in their swollen and hydrated state. However, as the temperature increases above LCST, these intermolecular bonds between the hydrophilic groups and the solvent weaken, giving rise to intramolecular bonds among hydrophobic segments. The polymer collapses, leading to a phase separation between the microgels and the surrounding medium, thus reducing its volume [9,10]. PNIPAAm can be crosslinked to create hydrogels or microgels that swell when in the presence of a suitable solvent, producing colloidal dispersions. This characteristic gives this polymer the high potential in biomedical applications such as drug release, biosensing, and tissue engineering [11,12,13,14,15].

Magnetic nanoparticles (NPs) are a class of inorganic materials widely used for biomedical applications in the last decades, namely for cancer theranostics applications [16,17,18,19,20]. The incorporation of magnetic nanoparticles into microgels gives rise to multi-responsive systems. These systems combine the thermosensitivity of the polymeric matrix and the magnetic properties of the nanoparticles, widening the field of potential applications and making them suitable for magnetic separation, drug delivery, magnetic resonance image contrast enhancement, and hyperthermia treatments [21,22,23,24]. The present work focuses on using these hybrid systems in magnetic hyperthermia applications. Superparamagnetic iron oxide nanoparticles (SPIONs) are especially interesting due to their biocompatibility and stability, being the most used magnetic NPs in biomedical applications. In magnetic hyperthermia, SPIONs can be localized at the tumor site and generate heat to reach therapeutic temperatures of around 41–45 °C, causing tumor cell death without harming healthy tissues [17]. Echeverria et al. [24] studied the development of PNIPAAm–chitosan–magnetic nanoparticle hybrid systems reporting a successful NPs encapsulation in the polymer matrix. The group obtained a temperature variation of 3.1 °C for PNIPAAm-CS20-10%IONPs, showing these hybrid systems’ potential in magnetic hyperthermia applications.

Microgels can be confined into polymeric fibers using the colloidal electrospinning technique, producing a flexible composite that responds to external stimuli and holds the characteristics of the electrospun fibers, such as a high surface area and high porosity, increasing the sensitivity of the system [25,26]. It also allows localized implantation of the dual-responsive system at the tumor site. Marques et al. [27] produced poly-(ethylene oxide) (PEO) fibers with PNIPAAm–chitosan (PNIPAAm–CS) microgels incorporated. The fibers had an average diameter of 63 nm and bead-on-a-string morphology. Polyvinylpyrrolidone (PVP) fibers confining PNIPAAm microgels were also reported. Díaz et al. [28] studied the absorption properties of composite fibers of (PNIPAAm)-based microgels using PVP as fiber template and could load up to 40 wt.% of microgels into the fibers. The group obtained fibers with a mean diameter of 0.9 µm and a bead-on-a-string morphology. On the same note, Faria et al. [25] also produced electrospun PVP-PNIPAAm composite membranes with a mean diameter of around 497 nm.

Electrospun fibers have unique characteristics that makes them especially interesting in the drug delivery field. Some of those characteristics are the high surface area of the fibers that allows efficient drug release, high porosity, and the vast possibility for surface functionalization. In addition to that, the simplicity and low cost of the electrospinning technique also allows convenient incorporation of a wide range of drugs within the fibers. The use of these devices for drug delivery could also reduce the systemic absorption of the drugs and offer a more localized treatment at lower concentrations [29,30]. Both the incorporation of PNIPAAm microgels and iron oxide nanoparticles in electrospun nanofibers address the issue of localization of these systems in tumor region in a sufficient concentration to enable effective treatment [31]. Also, the high surface area and porosity of the nanofibers allow the increase of the sensitivity of the system and consequently, its efficacy [25]. The present work aims to develop and characterize dual-stimuli responsive PNIPAAm-NPs microgels and their incorporation in electrospun polymeric fibers through colloidal electrospinning for biomedical applications such as magnetic hyperthermia. PVP was used as a fiber template and adequately cross-linked to be used in physiological conditions without membrane dissolution. Composite membranes were fully characterized and their mechanical properties, swelling ability, and heating ability were evaluated.

## 2. Results and Discussion

The magnetic component of the dual-stimuli responsive device is composed of SPIONs. Superparamagnetic magnetite (Fe_3_O_4_) NPs were extensively studied by our research group previously [17,32,33]. To improve the stability of iron oxide NPs in aqueous environment, two surfactants were used: Oleic acid (OA) and dimercaptosuccinic acid (DMSA). In previous work, it was demonstrated that monodisperse bare Fe_3_O_4_ NPs with an average diameter of 9.4 ± 1.9 nm (measured in transmission electron microscopy images—TEM) were successfully stabilized with OA and DMSA without significantly changing the core diameter (9.8 ± 2.5 nm and 10.2 ± 2.4 nm, respectively). Additionally, X-ray diffractograms showed the presence of magnetite in all samples with the negligible effect of the surfactants in the crystalline structure of Fe_3_O_4_ NPs. Finally, infrared spectra indicated the presence of OA and DMSA at Fe_3_O_4_ NPs surface [32].

### 2.1. Production of Magnetic PNIPAAm Microgels

The thermoresponsive component of the dual-stimuli responsive device is composed of PNIPAAm microgels. In our previous work, PNIPAAm microgels with various contents of chitosan and Fe_3_O_4_ NPs were produced using the surfactant-free emulsion polymerization (SFEP) technique [24]. Using the same technique, we produced plain PNIPAAm microgels, and PNIPAAm-CS microgels with Fe_3_O_4_ NPs incorporated. In the latter, bare Fe_3_O_4_ NPs and Fe_3_O_4_ NPs coated with OA or DMSA were used to evaluate the surfactant’s effect on the incorporation efficiency.

Fourier transform infrared (FTIR) measurements were firstly performed in all samples to evaluate their chemistry. Figure 1A shows the FTIR spectrum of PNIPAAm microgels and the monomer used in its synthesis, NIPAAm. In the PNIPAAm spectrum, it is possible to identify the band at 3290 cm^−1^ that corresponds to the -NH stretching vibration. The absorption region between 2800 and 3000 cm^−1^ is due to C–H stretching vibrations of –CH_3_ and –CH_2_– groups [34,35]. The band vibration amide I at 1630 cm^−1^ is assigned to the stretching of the C=O bond, while amide II vibration at 1520 cm^−1^ corresponds to the stretching of -NH bond. The band’s disappearance at 1620 cm^−1^ in the PNIPAAm spectrum shows the breaking of C=C bond present in the monomer spectrum. This confirms the formation of a long chain polymer backbone due to the free radical polymerization [36]. FTIR spectra of PNIPAAm microgels with Fe_3_O_4_ NPs incorporated (Figure 1B) show the presence of PNIPAAm typical bands with the addition of some characteristic bands of chitosan. The band at 3330 cm^−1^ corresponds to the elongation of –OH and –NH bonds, while at 2920 cm^−1^ and 2860 cm^−1^, two small bands correspond to the asymmetric vibration of –CH_3_ and –CH_2_ from the chitosan backbone. Additionally, the band at 1660 cm^−1^ overlaps with the same band from PNIPAAm microgels which correspond to –NH stretching vibration, while the band at 1060 cm^−1^ corresponds to the anti-symmetric stretching of C–O–C bridge [37]. Finally, spectra of microgels with the addition of Fe_3_O_4_ NPs also present the characteristic band of Fe–O at 560 cm^−1^ [33].

Thermogravimetric analysis (TGA) of all samples was performed to evaluate the thermal stability of PNIPAAm microgels with CS and Fe_3_O_4_ NPs incorporated. Indirectly it is also possible to assess the presence of Fe_3_O_4_ NPs in the samples. Figure 1C,D shows the TGA and corresponding differential thermal analysis (DTA) curves of all MG samples, respectively. In all cases, the first step of thermal degradation is present from 25 °C to 120 °C, which is attributed to water content. In the PNIPAAm MG sample, the second step of thermal degradation is present from 180 °C to 450 °C, which is related to PNIPAAm decomposition [38]. When chitosan is incorporated (MG-CS) an additional degradation step with a maximum peak of around 310 °C appears. This degradation step is attributed to chitosan degradation, including the dehydration of saccharide rings, depolymerization, and decomposition of the polymer backbone [37]. Additionally, it is noteworthy that PNIPAAm has stronger thermal stability than chitosan. When Fe_3_O_4_ NPs are incorporated into MG, the degradation steps occur in similar temperatures as in plain PNIPAAm microgels, indicating that Fe_3_O_4_ NPs do not affect the thermal stability of PNIPAAm microgels.

Nevertheless, the residual mass increases, which can be attributed to the presence of Fe_3_O_4_ NPs. As reported elsewhere, iron oxide NPs show only a small decrease in mass percentage up to 900 °C which is attributed to the conversion of Fe_3_O_4_ to γ-Fe_2_O_3_ and FeO above 570 °C [16]. Comparing the residual mass of the three samples with Fe_3_O_4_ NPs (without coating, and with OA or DMSA) it is visible that the residual mass is smaller when Fe_3_O_4_-OA NPs are incorporated, comparing with Fe_3_O_4_-DMSA and bare Fe_3_O_4_ NPs. This can be an indication that the incorporation of Fe_3_O_4_ NPs could be compromised by the presence of OA.

Morphological analysis of PNIPAAm microgels was performed by means of TEM (Figure 2). Monodisperse PNIPAAm microgels with spherical shape and an average diameter of 656 nm were produced. The incorporation of Fe_3_O_4_ NPs decreases the average diameter of PNIPAAm microgels to values from 200 to 400 nm, depending on the type of Fe_3_O_4_ NPs incorporated, which correlates with our previous work [24]. Analyzing PNIPAAm microgels with bare Fe_3_O_4_ NPs incorporated (Figure 2B), it is visible that Fe_3_O_4_ NPs are located preferably at the microgel surface rather than inside the microgel. This preferable localization can be explained by the presence of chitosan, which acts as a surfactant to stabilize Fe_3_O_4_ NPs and PNIPAAm microgels, thus being preferably located at the microgel surface [24]. When Fe_3_O_4_-OA NPs are incorporated into PNIPAAm microgels (Figure 2C), phase separation occurred, which indicates that a large part of the Fe_3_O_4_ NPs is not successfully incorporated, as seen in TGA results. TEM image of PNIPAAm microgels with Fe_3_O_4_-DMSA NPs (Figure 2D) show that these Fe_3_O_4_ NPs are dispersed inside the microgel, producing microgels with an average diameter of 185 nm. Although both oleic acid and DMSA have carboxylic groups, their pka is different, influencing their respective incorporation. DMSA pka is 3.37, which is lower than the pH of the polymerization reaction (around 5.5), meaning that the carboxylic groups are in their ionization state. In this state, these negatively charged groups easily interact with PNIPAAm backbone structure, thus improving the incorporation of Fe_3_O_4_-DMSA NPs. On the other hand, oleic acid pka is 5.0, translating into a smaller amount of ionized carboxylic groups at the reaction pH [16]. That said, the incorporation of Fe_3_O_4_-OA NPs into PNIPAAm microgels occurs to a lower degree, as seen in TEM images and TGA results. Figure 3 depicts the schematic representation of the produced components (PNIPAAm microgels and Fe_3_O_4_ NPs) in order to better understand the interactions between them.

Dynamic light scattering (DLS) was used to determine the colloidal stability of the produced microgels and their hydrodynamic diameter as a function of temperature. Figure 4A depicts the evolution of the hydrodynamic diameter with temperature for PNIPAAm microgels. A decrease in the hydrodynamic diameter is evident with the increase of temperature, which agrees with the negative responsive behavior typically demonstrated by this polymer. From these data, a VPTT of 31 °C was determined according to what is reported in the literature [24]. PNIPAAm microgels are in a swollen state at room temperature with a hydrodynamic diameter of around 900 nm. Above a critical temperature (in this case 31 °C), the hydrodynamic diameter decreases, and at higher temperatures, the collapsed state of the microgels takes place, and their diameter remains constant, around 400 nm. The same behavior is observed in the autocorrelation curves obtained from DLS measurements (Figure 4B). These curves show a typical single-exponential relaxation, with a smaller decay time as temperature increases. This behavior is related to PNIPAAm microgels collapse as the temperature increases above LCST. As microgels start to collapse by reaching their intrinsic LCST, their hydrodynamic diameter decreases, translating into a shorter decay time in DLS measurements [39].

The same analysis was performed for PNIPAAm-CS and hybrid microgels with Fe_3_O_4_ NPs (Figure 4C). Additionally, the volume phase transition was evaluated by normalizing the obtained diameters in DLS. To do that, the relative swelling of each sample, (d/d_0_)^3^, was calculated by normalizing the respecting diameter (d) by the reference diameter value of the microgel in its collapsed state (d_0_) (Figure 4D). It is visible that the incorporation of chitosan into PNIPAAm microgels does not affect the LCST. However, it dramatically decreases the hydrodynamic diameter of the microgels below their critical temperature. From these data, it is possible to infer that chitosan may act as a crosslinking agent, causing the decrease of microgels hydrodynamic diameter. Jaiswal et al. [36] observed this phenomenon in PNIPAAm-CS based nanohydrogels incorporated with Fe_3_O_4_ magnetic nanoparticles. Average particle size and hydrodynamic diameter of the nanogels decreased with the incorporation of chitosan and continue decreasing with a higher concentration of chitosan present in the nanogels. These results are also consistent with those reported by Echeverria et al. [24] that also studied the influence of chitosan in PNIPAAm microgels. This also explains the decrease in the swelling ability of the MG-CS microgels since the present of a crosslinker decreases the mobility of PNIPAAm chains, limiting water uptake.

Additionally, some scattered points are visible at higher temperatures. This phenomenon can be explained by the release of chitosan molecules when microgels collapse, resulting in the presence of large agglomerates as the temperature increases [24]. Samples containing bare Fe_3_O_4_ NPs and Fe_3_O_4_-DMSA NPs demonstrate that incorporating these NPs decreases the hydrodynamic diameter of PNIPAAm microgels. However, the relative swelling below LCST is higher compared to plain PNIPAAm microgels. However, when Fe_3_O_4_-OA NPs are incorporated, a similar behavior to PNIPAAm-CS sample is observed. As previously discussed, Fe_3_O_4_-OA NPs are not effectively contained inside the microgel, leading to a predominant effect of chitosan in PNIPAAm microgel diameter and swelling. Nevertheless, in all cases, LCST of PNIPAAm microgels is not affected by incorporating chitosan or Fe_3_O_4_ NPs.

### 2.2. Production of Magnetic Membranes via Colloidal Electrospinning

Following this research paper’s primary purpose, i.e., to produce a dual-stimuli responsive device, PNIPAAm was incorporated into a polymeric matrix via colloidal electrospinning. To produce this matrix, the electrospinning technique was chosen as a simple, low cost and effective method to produce flexible nanofibrous membranes that can be easily implanted [32]. Firstly, plain PNIPAAm microgels were incorporated into polymeric fibers composed of PVP to optimize their encapsulation through colloidal electrospinning. Secondly, magnetic PNIPAAm microgels were incorporated into PVP electrospun fibers. Finally, dual-stimuli responsive membranes were extensively characterized, and their behavior in magnetic hyperthermia application was assessed.

Electrospinning parameters to produce PVP membranes were optimized. PVP membranes were obtained using a 14 wt.% PVP solution in water/ethanol (35:65) electrospun with a voltage of 20 kV, a tip-to-collector distance of 20 cm at a flow rate of 0.3 mL·h^−1^. Figure 5A shows SEM image of PVP membrane obtained with the abovementioned conditions, demonstrating a uniform membrane composed of fibers without beads and an average diameter of 420 ± 53 nm. When 10 wt.% of PNIPAAm microgels are incorporated in PVP solution (Figure 5B,C), a slight increase of the resultant fibers’ diameter occurred (456 ± 78 nm). However, the presence of MG cannot be confirmed by SEM, because the morphology of PVP fibers did not change. This occurs because, in their collapsed state, PNIPAAm microgels have diameters smaller than the fiber diameter. Therefore, to confirm the presence of magnetic MG inside the fibers, TEM was performed (Figure 5C). Despite the low contrast obtained due to the materials’ nature (organic materials), it is possible to see microgels inside PVP fibers. Similar morphology was observed in previous work incorporating PNIPAAm-AAc microgels into PVP fibers [25].

Additionally, when magnetic MG were incorporated (Figure 5D–F), higher contrast is obtained due to the inorganic phase (Fe_3_O_4_ NPs) inside PNIPAAm microgels, confirming the successful incorporation. No significant differences were found with incorporating hybrid MG with Fe_3_O_4_ NPs coated with DMSA or OA. In both cases, a homogeneous distribution of the microgels throughout the fibers was observed.

### 2.3. Mechanical and Swelling Behaviour of Magnetic Composites

After morphology analysis, the mechanical and swelling behavior of composite membranes was analyzed. Through the study of the stress curves, it was possible to extrapolate the value of Young’s modulus (*E*), which characterizes the mechanical resistance of the material, and the ultimate tensile strength (UTS), which corresponds to the maximum stress value that the material can withstand without fracturing. Figure 6A,B show these parameters for the different produced membranes. Non-crosslinked and crosslinked PVP membranes show Young’s modulus of around 20 and 31 MPa, respectively. This indicates that crosslinking increases membrane stiffness. It is also visible an increase in the UTS of the membranes due to its crosslinking. The incorporation of PNIPAAm microgels in PVP membranes improved their mechanical parameters compared to the plain crosslinked membrane. Also, all composite samples exhibited a similar mechanical behavior under stress tests showing a major elastic deformation region and a negligible plastic deformation region, behavior characteristic of fragile materials. This demonstrates that the incorporation PNIPAAm microgels have a reinforcement effect on PVP membranes.

The swelling ratio of the membranes was obtained by measuring their mass before and after immersion. Finally, the swelling ratio was calculated through Equation (1).
(1)$$Q=\frac{w_{s}-w_{d}}{w_{d}}$$
where *w_s_* is the weight of the membrane after immersion and *w_d_* the weight of the membrane before immersion. Figure 6C shows the swelling curves for plain and composite PVP membranes. By analyzing the swelling ratio curves, it is possible to verify that plain PVP reaches the highest swelling ratio, of approximately 10.2 (g/g) after 4 h of immersion. It is also possible to observe that PVP membranes with PNIPAAm microgels incorporated have a lower swelling ratio of approximately 8.6 (g/g). This decrease in the swelling ratio could be explained due to the presence of the incorporated microgels. Faria et al. [25] explain this phenomenon with the diffusion path of water in the polymeric fibers. When a fluid penetrates a network, it chooses the path of least resistance, i.e., a straight line. In composite membranes, the incorporation of microgels creates obstacles that interfere with the normal diffusion of water, forcing it to work around the obstacles. This means that the average path that water traveled in the membranes with microgels is longer than the path traveled in plain membranes. The ratio between the actual distance (Δt) travelled by the species per unit of length of the medium (Δx) is called tortuosity (τ) (Equation (2)) [40].
(2)$$\tau =\frac{\Delta t}{\Delta x}$$

This means that membranes incorporated with microgels have a higher tortuosity than plain membranes. This parameter has a high influence in the diffusion coefficient of a fluid through a membrane and their relation can be shown in Equation (3).
(3)$${d^{\prime }}_{m}=\frac{d_{m}}{\tau^{2}}$$where *d*’*_m_* is the diffusion coefficient, dm is the diffusion coefficient of the species without the presence of ‘obstacles’ (in this case, microgels) and τ the tortuosity. As it can be seen in Equation (3), the tortuosity is inversely proportional to the diffusion coefficient, meaning that PVP MG membranes have a lower diffusion coefficient and therefore swell less than plain PVP membranes.

The presence of chitosan in the microgels (PVP MG-CS) incorporated in the membranes decreases the swelling ratio to around 7 (g/g). As described in DLS tests, chitosan may act as a crosslinking agent, reducing the swelling index of microgels and, consequently, membranes’ swelling capacity. Finally, the incorporation of nanoparticles in the microgels present in the electrospun membranes does not significantly influence their swelling ratio since, considering the associated error, this swelling value does not vary much from that of PVP MG-CS membranes. Nevertheless, membranes containing MG with bare Fe_3_O_4_ NPs have a higher swelling ratio than membranes with MG-OA or MG-DMSA. In MG-OA samples, the low encapsulation efficiency can lead to the presence of free Fe_3_O_4_-OA NPs in PVP fiber, which can also act as obstacles for water migration, like PNIPAAm microgels. In MG-DMSA samples, the homogeneous distribution of Fe_3_O_4_-NPs throughout the microgel can influence its swelling ability, and consequently, the composite membrane swelling ratio.

Through the analysis of the swelling behavior of the membranes, it is also possible to calculate a series of parameters directly related to the crosslinking degree of the polymer network: molecular weight of the polymer chain between two neighboring crosslinking nodes ($\bar{M_{c}})$, crosslinking density of the network (ρ_x_) and mesh size (ξ). From swelling data, $\bar{M_{c}}$ can be obtained through applying the Equilibrium Swelling theory (Equation (4)), assuming that the chemical crosslinking gives rise to the formation of a hydrogel [41].
(4)$$\frac{1}{\bar{M_{c}}}=\frac{2}{\bar{M_{n}}}-\frac{\upsilon /V_{1}\left[\ln\left(1-\upsilon_{2,s}\right)+\upsilon_{2,s}+\chi_{1}\upsilon_{2,s}^{2}\right]}{\upsilon_{2,r}\left[{\left(\frac{\upsilon_{2,s}}{\upsilon_{2,r}}\right)}^{\frac{1}{3}}-\left(\frac{\upsilon_{2,s}}{\upsilon_{2,r}}\right)\right]}$$
where $\bar{M_{n}}$ represents PVP average molecular weight of before crosslinking, 1.300 kDa, υ is the specific volume of the polymer (0.785 cm^3^·g^−1^), $V_{1}$ represents the molar volume of water (18.1 cm^3^·mol^−1^), $\upsilon_{2,r}$ designates the polymer volume fraction of the crosslinked PVP in the relaxed state, $\upsilon_{2,s}$ is the polymer volume fraction of the PVP in the swollen state and χ denotes the Flory PVP polymer–solvent interaction parameter, 0.48 for PVP/water. The mesh size of the crosslinked network (ξ) is the linear distance among successive crosslinks. It represents the space available for the diffusion of solute and can be calculated with the help of Equation (5).
(5)$$\xi =\;\upsilon_{2,s}^{-\frac{1}{3}}l{\left(\frac{2C_{n}\bar{M_{c}}}{M_{r}}\right)}^{\frac{1}{2}}$$
where l is the length of the bond along the polymer’s backbone (1.54 Å), $C_{n}$ is the Flory characteristic ratio (12.3) and $M_{r}$ represents the molecular weight of the repeating units of PVP (111.14 g·mol^−1^) [42,43]. Finally, the crosslinking density of the network can be determined through the following equation:(6)$$\rho_{x}=\frac{1}{\upsilon \bar{M_{c}}}.$$

Table 1 shows the variation of $\bar{M_{c}}$, ρ_x_ and  ξ between bare and hybrid PVP membranes. Figure 6D shows the mesh size (ξ) and crosslinking degree (ρ_x_) of the produced PVP membranes without and with PNIPAAm microgels incorporated. It is visible that the mesh size and crosslinking density are inversely proportional, which means that if the crosslinking density of the network is higher, the distance between successive crosslinks decreases. The incorporation of plain PNIPAAm microgels into PVP membranes increases the crosslinking degree and reduces mesh size. A higher crosslinking density in the membrane means a decrease in the polymer chains’ mobility, which results in a lower water uptake, thus making the plain membranes have a higher swelling ratio than the composite ones. This increase in crosslinking degree can result from PNIPAAm microgels presence as an obstacle for water to penetrate through PVP fiber. Since both values are determined from the swelling ratio, a decrease in the swelling ratio will increase the crosslinking degree, which may not be associated with a higher crosslinking density.

Furthermore, PNIPAAm microgels can act as anchoring points in PVP fiber that affect both the mechanical properties and the swelling ratio. As previously described, it was observed that composite PVP MG membranes present higher values of Young’s Modulus and UTS. These composite membranes also show a higher crosslinking density, which means that the force that is applied per unit of area and the necessary force required to rupture the membrane is higher. That said, composite membranes present a higher Young’s Modulus and UTS than plain PVP membranes.

The incorporation of hybrid microgels into PVP membranes have a higher effect on crosslinking degree and mesh size, particularly microgels containing Fe_3_O_4_-OA and Fe_3_O_4_-DMSA NPs. As previously reported, the incorporation of Fe_3_O_4_ NPs in PNIPAAm microgels can act as crosslinking points, thus decreasing their relative swelling index. Therefore, if microgels contribute to PVP composite membranes swelling ratio, microgels with a smaller swelling index led to a smaller swelling ratio of the composite membrane. These values translate into a higher crosslinking degree and smaller mesh size. Additionally, in MG-CS and MG-OA samples, the instability caused by chitosan release from the microgel (as seen in DLS results) can lead to a higher effect in the swelling ratio. In MG-DMSA, the fact that Fe_3_O_4_ NPs are homogeneously dispersed throughout the whole microgel structure can significantly influence PNIPAAm microgel’s interaction with PVP fiber, thus increasing the later crosslinking degree.

### 2.4. Magnetic Hyperthermia Assays

Magnetic hyperthermia assays were performed to evaluate the heating ability of Fe_3_O_4_ NPs incorporated in PNIPAAm microgels and composite PVP membranes. Figure 7A represents the heating ability of the PVP membranes with hybrid PNIPAAm-NPs microgels incorporated and the heating ability of the theoretical amount of MG-NP and Fe_3_O_4_ NPs contained in the membranes. By analyzing the hyperthermia results shown in Figure 7 it is possible to verify that bare Fe_3_O_4_ NPs appear to have a higher temperature variation than coated NPs (OA or DMSA). This may be due to the stabilizing agent forms a layer around the nanoparticles, which can restrict their Brownian movements when subjected to an alternating magnetic field and restrict their ability to generate heat [10]. For all three systems, the temperature variation for the same theoretical concentrations of NPs is similar, indicating the excellent incorporation of nanoparticles in the microgels and microgels in the PVP membranes. The highest temperature variation in the PVP composite membranes occurred for those incorporated with bare Fe_3_O_4_ NPs, resulting in an average temperature variation of 1 °C. Figure 7B presents the specific absorption rate (SAR) for all samples. SAR measures the heating efficiency of magnetic nanoparticles through energy absorption when an alternating magnetic field is applied. The value is determined by Equation (7):(7)$$SAR\;\left(W/g\right)=\frac{C_{NP}m_{Fe}+C_{l}m_{l}}{m_{Fe}}{\left(\frac{dT}{dt}\right)}_{max}.$$where (*dT*/*dt*)*_max_* is the maximum derivative of the temperature curve, *C_NP_* is the specific heat of Fe_3_O_4_ NPs, *C_l_* is the specific heat of the liquid, m_l_ is the fluid mass, and *m_Fe_* is the iron mass in the sample. As expected from temperature variation results, SAR values are similar between all samples. With the incorporation of bare Fe_3_O_4_ NPs in microgels and PVP membrane, a small decrease of SAR value is observed due to the limitation in Brownian relaxation. The same is observed in Fe_3_O_4_-OA samples, where OA is responsible for decreasing the generated heat by Brownian relaxation, being Néel relaxation the predominant mechanism [24]. Fe_3_O_4_-DMSA NPs show a higher SAR value, which may be related to better stabilization of DMSA than OA, and consequently, fewer restrictions in NPs Brownian relaxation.

### 2.5. Cytotoxicity Assays

To evaluate the cytotoxicity of magnetic colloidal membranes, the various formulations of the membranes were exposed to culture medium for 24 h. These extracts were then exposed to Vero cells and cell viability was assessed after 48 h. Figure 7C shows the obtained results for cell viability after 24 h of exposure to the different types of produced membranes. The results are expressed in % of cell viability calculated by [% cell viability = NP treated cells/control cells × 100]. In all cases, an initial concentration (40 mg of sample per mL) and three dilutions of factor 2 were performed. Cytotoxicity assays were performed to evaluate the cytotoxic effect of PNIPAAm microgels and Fe_3_O_4_ NPs incorporated in PVP membranes. All assays reveal that composite membranes do not present any cytotoxicity and therefore can be used in biomedical applications.

## 3. Conclusions

In this work, a dual-stimuli responsive system was developed composed of magnetic nanoparticles incorporated into thermoresponsive microgels, which in turn were incorporated into PVP electrospun membranes. Effective incorporation of magnetic nanoparticles into PNIPPAm microgels was demonstrated without significantly affect the thermoresponsiveness of the microgels. Fe_3_O_4_ NPs coated with DMSA demonstrated better incorporation efficiency, being homogeneously distributed throughout the microgel. Additionally, hybrid MG were successfully incorporated into PVP membranes without affecting their morphology. The mechanical properties of composite membranes improved, increasing their elasticity without significantly affect their resistance at the break. This can also be explained by hybrid microgels act as an anchoring point in PVP fibers, thus increasing their crosslinking degree, thus decreasing their swelling ratio. The final composite membranes did not show any toxicity to Vero cells under the tested experimental conditions. Additionally, magnetic hyperthermia results demonstrated that incorporating Fe_3_O_4_ NPs into microgels and PVP membranes does not significantly affect their heating ability, resulting in a temperature variation adequate for cancer treatment through magnetic hyperthermia.

This work demonstrated the development of a new dual-stimuli responsive system with a high potential for magnetic hyperthermia application. Using colloidal electrospinning, it is possible to easily incorporate hybrid microgels into a scaffold to produce multifunctional devices. Combining this technique with stimuli-responsive materials make this an exciting and promising research topic. The developed system also present a huge potential to produce advanced drug delivery systems that can be remotely triggered by an external stimulus like temperature or magnetic field.

## 4. Materials and Methods

### 4.1. Synthesis of Iron Oxide Nanoparticles

Iron oxide NPs were synthesized by chemical co-precipitation as described by Soares et al. [33]. Iron chloride tetrahydrate (FeCl_2_·4H_2_O, 2.5 mmol, Alfa Aesar, Ward Hill, MA, USA) and iron chloride hexahydrate (FeCl_3_·6H_2_O, 5 mmol, Alfa Aesar) were mixed in ultrapure water (Millipore, Burlington, MA, USA) followed by the addition of an ammonium solution (NH_4_OH at 25%, 10 mL, Panreac, Chicago IL, USA) to precipitate the IONPs in the absence of oxygen, by bubbling N_2_.

Iron content in iron oxide nanoparticles was determined using the 1,10-phenanthroline colorimetric method [43] to calculate the amount of dimercaptosuccinic acid (DMSA, Acros Organics, Geel, Belgium, 98%) and oleic acid (OA, Fisher Chemical, Fair Lawn, NJ, USA) to add to the magnetic nanoparticles. The mixture reacted for 3 h in an ultrasound bath. After the reaction, the stabilized NPs were subjected to dialysis in a 4 RC Dialysis Membrane Tubing 12 to 14 kDa MWCO (SpectrumTM Spectra/PorTM) to remove the excess surfactant.

### 4.2. Depolymerization of Chitosan

Chitosan ((C_8_H_13_NO_5_)_n_), 469 kDa, Cognis, Monheim, Germany) was depolymerized by a chemical reaction with sodium nitrite (NaNO_2_, Sigma-Aldrich, St. Louis, MO, USA). Briefly, 2.5 g of chitosan was dissolved in 250 mL of acetic acid 1% (*v*/*v*) (Fisher Chemical). After complete dissolution, the desired amount of sodium nitrite solution was added to the chitosan solution and mechanically stirred for 1 h at 1000 rpm. 1 M sodium hydroxide (NaOH, EKA, Karnataka, India) solution was added to the reaction mixture dropwise to precipitate low molecular weight chitosan. The resulting suspension was centrifuged at 10,000 rpm for 10 min, washed several times with ultrapure water, freeze-dried, and stored in a dry place.

### 4.3. Synthesis of PNIPAAm Microgels

PNIPAAm microgels were synthesized by surfactant-free emulsion polymerization (SFEP) described by Echeverria et al. [24]. *N*-Isopropylacrylamide (NIPAAm, Aldrich Chemistry, St. Louis, MO, USA, 97%) was used as monomer and *N*,*N*-methylenebisacrylamide (MBA, Sigma-Aldrich, 99%) as a crosslinker. Ammonium persulfate (APS, Sigma-Aldrich, 99%) and sodium bisulfite (SBS, Acros Organics) were used respectively as the initiator and the reaction’s catalyzer. The polymerization reaction was conducted at 70 °C in a 250 mL three-necked round bottom flask under a nitrogen atmosphere. NIPAAm/water solution was added to the round bottom flask that was subsequently stirred and purged with nitrogen during 30 min. MBA/water and SBS/water solutions were added to the reaction vessel, and the temperature was raised up to 70 °C. The reaction was thermally initiated by adding APS/water solution with a total water amount of 100 mL. The polymerization reaction continued under nitrogen atmosphere at a constant mechanical stirring of 450 rpm. After 5 h, the reaction was terminated by cooling down to room temperature, and the obtained samples were dialyzed for one week.

For the preparation of hybrid microgels, 40 wt.% of chitosan and 5 wt.% of magnetic nanoparticles were added to the polymerization synthesis. Firstly, depolymerized chitosan was previously dissolved in 50 mL of 1% (*v*/*v*) acetic acid solution for 24 h. The dissolved chitosan and the nanoparticles solution were added to the reactor before following the previously mentioned polymerization process methodology.

### 4.4. Electrospinning Process

Polyvinylpyrrolidone 14 wt.% solution (PVP, Sigma-Aldrich, Mw = 1,300,000 Da) was used as a fiber template. The solutions were prepared by dissolving the desired amount of PVP into water and ethanol (Honeywell, Charlotte, NC, USA, 99%) in a 35:65 ratio so that the co-nonsolvency of the microgels was ensured [44] and that the microgels were confined in their swollen state. The solution was magnetically stirred for 3 h at room temperature to ensure complete homogenization.

The solutions were loaded into a 1 mL syringe with a 23-gauge blunt tip needle and mounted onto a syringe pump programmed with a flow of 0.1 and 0.3 mL·h^−1^. The process was carried out with two different applied voltages, 15 and 20 kV, and two different tip-to-collector distances (TCD), 15 and 20 cm. The study of the optimal electrospinning parameters was carried out to obtain fibers with the smallest average diameter possible. After optimizing the electrospinning process, PNIPAAm microgels were added to PVP solutions to achieve 10 wt.%, regarding PVP total mass. The produced membranes were crosslinked at 165 °C for 24 h [45].

### 4.5. Characterization

Fiber morphology and diameter were observed by scanning electron microscopy (SEM, Carl Zeiss Auriga, Jena, Germany). The samples were coated with a thin layer of gold. Transmission electron microscopy (TEM) images were obtained using a Hitachi H-8100 II equipment with thermionic emission LaB6.

FTIR spectrum of the samples was obtained with the Thermo Nicolet 6700 FTIR spectrophotometer. The spectrum was acquired with a 45° incident angle in the range of 500–4500 cm^−1^, with a 2 cm^−1^ resolution.

The thermal properties of the samples were analyzed by thermogravimetric analysis technique using the NETZSCH STA 449F3 equipment, in a temperature range between 25 and 900 °C with a heating rate of 10 °C·min^−1^, under a flow of nitrogen gas.

Dynamic light scattering (DLS) measurements were performed using a Horiba SZ-100 Nanopartica Analyzer light scattering instrument with a 532 nm laser to determine the hydrodynamic diameter of microgels and their swelling behavior in aqueous medium, in the range of temperatures from 25 to 40 °C.

Tensile tests were performed to study the membranes mechanical response. The samples were cut into 20 × 10 mm rectangles and stretched at a speed of 1 mm·min^−1^. The assays were performed on a 20 N load cell Rheometric Scientific uniaxial machine operable with the “Minimat” software (Minimat Control Software Version 1.60 February 1994 (c) P.L. Thermal Science 1984-94 Rheometric Scientific Ltd.).

Swelling behavior of PVP crosslinked membranes was studied at pH 7.4 at 37 °C. Membranes were cut in 1 × 1 cm squares and immersed in a phosphate buffer solution (PBS) for different amounts of time, 5, 10, and 30 min and 1, 2, and 4 h. Samples were weighed before and after their immersion in PBS to assess mass variation.

Magnetic hyperthermia measurements were obtained with the NanoScale Biomagnetics equipment, DM100 Series working at a frequency of 418.5 kHz and field intensity of 24 kA·m^−1^ during 10 min.

### 4.6. Cytotoxicity Assay

To evaluate the cytotoxicity of the membranes, the assays were performed according to standard ISO-10993 Biological evaluation of medical devices, Part 5: Tests for in vitro cytotoxicity. The assays were performed using the extract method and Vero cells (monkey renal epithelial cells). To produce the extract, different samples were cut to have an identical mass of 80 mg and pre-sterilized with UV irradiation for 1 h. Each sample was placed in 2 mL of Dulbecco’s modified Eagle’s medium (DMEM, Sigma Aldrich) at 37 °C for 24 h. Four concentrations of the extract were used 40, 20, 10, and 5 mg·mL^−1^. For each concentration, 5 replicas were performed. Cells were seeded at a density of 20,000 cell cm^−2^ in 96-well plates and grown in DMEM supplemented with 10% fetal bovine serum, 1% Penicilin-Streptomycin, sodium pyruvate (100 mM, Life-Technologies), and GlutaMAX™ Supplement (Life-Technologies, Carlsbad, CA, USA) followed by incubation at 37 °C in 5% CO_2_ during 48 h. Negative control cells were incubated with complete medium. Positive control cells were treated with 10% DMSO to cause cell death. After this period, the medium was removed and a resazurin solution containing 90% of complete culture medium and 10% of a 0.2 mg·mL^−1^ resazurin solution in 1× PBS was added to each well. After 2 h incubation, the absorbance was measured at 570 and 600 nm. Cell viability is expressed as a percentage of the negative control, given by [% cell viability = treated cells/control cells × 100].

## Figures and Tables

**Figure 1 gels-07-00028-f001:**
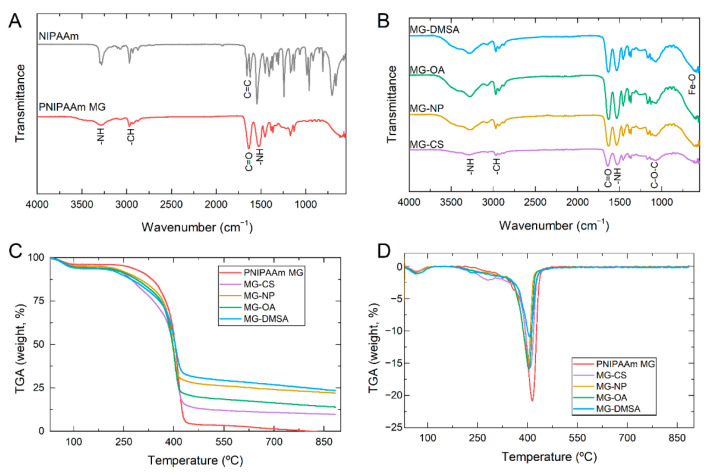
Fourier transform infrared (FTIR) spectra of Poly (N-isopropyl acrylamide) microgels (PNIPAAm MG) compared to its monomer, *N*-isopropyl acrylamide (NIPAAm) (**A**), and PNIPAAm microgels with chitosan (CS) and Fe_3_O_4_ NPs incorporated (**B**). Thermogravimetric analysis (**C**) and the corresponding derivative (DTA) (**D**) of plain PNIPAAm microgels, and PNIPAAm microgels with chitosan and Fe_3_O_4_ NPs incorporated. (MG-CS: PNIPAAm microgels with chitosan; MG-NP: PNIPAAm microgels with bare Fe_3_O_4_ NPs; MG-OA: PNIPAAm microgels with Fe_3_O_4_ NPs coated with oleic acid; MG-DMSA: PNIPAAm microgels with Fe_3_O_4_ NPs coated with dimercaptosuccinic acid).

**Figure 2 gels-07-00028-f002:**
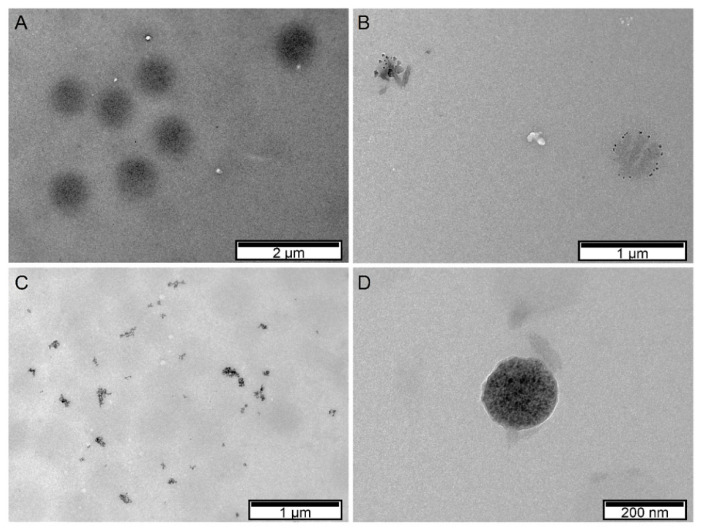
TEM images of bare PNIPAAm microgels (**A**), hybrid PNIPAAM microgels incorporated with bare Fe_3_O_4_ nanoparticles (NPs) (**B**), with Fe_3_O_4_-OA NPs (**C**), and with Fe_3_O_4_-DMSA NPs (**D**). (OA: Oleic acid; DMSA: Dimercaptosuccinic acid).

**Figure 3 gels-07-00028-f003:**
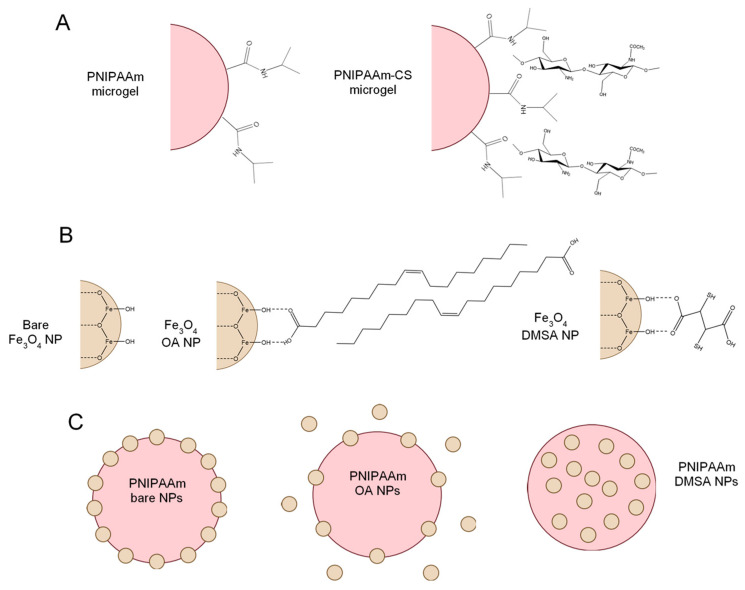
Schematic representation of the produced PNIPAAm and PNIPAAm-CS microgels (**A**) bare Fe_3_O_4_ NPs, Fe_3_O_4_-OA NPs, and Fe_3_O_4_-DMSA NPs (**B**) and hybrid PNIPAAm microgels incorporated with the different types of NPs (**C**).

**Figure 4 gels-07-00028-f004:**
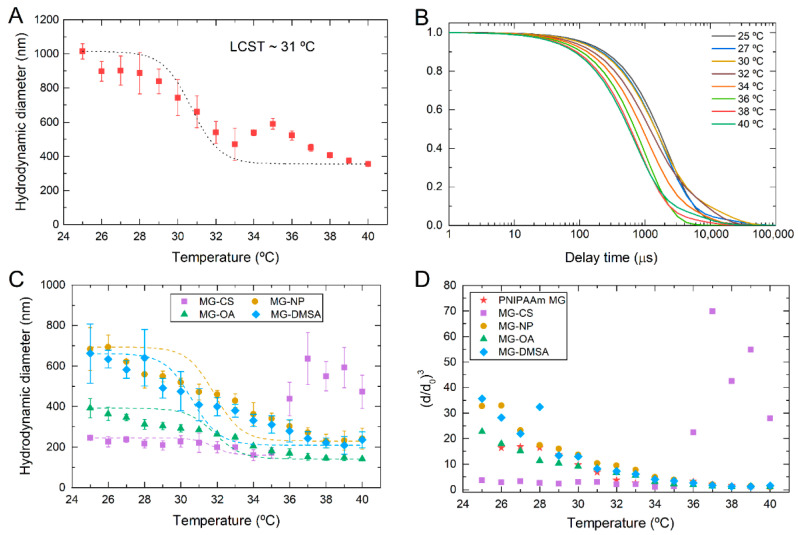
(**A**) Hydrodynamic diameter (D_H_) of PNIPAAm microgels as a function of temperature, and the corresponding autocorrelation curves (**B**), denoting a Lower Critical Solution Temperature (LCST) around 31 °C. (**C**) Hydrodynamic diameter of composite PNIPAAm microgels as a function of temperature; (**D**) Relative swelling index ((d/d_0_)^3^) as a function of temperature of plain PNIPAAm microgels and composite microgels. (MG-CS: PNIPAAm microgels with chitosan; MG-NP: PNIPAAm microgels with bare Fe_3_O_4_ NPs; MG-OA: PNIPAAm microgels with Fe_3_O_4_ NPs coated with oleic acid; MG-DMSA: PNIPAAm microgels with Fe_3_O_4_ NPs coated with dimercaptosuccinic acid).

**Figure 5 gels-07-00028-f005:**
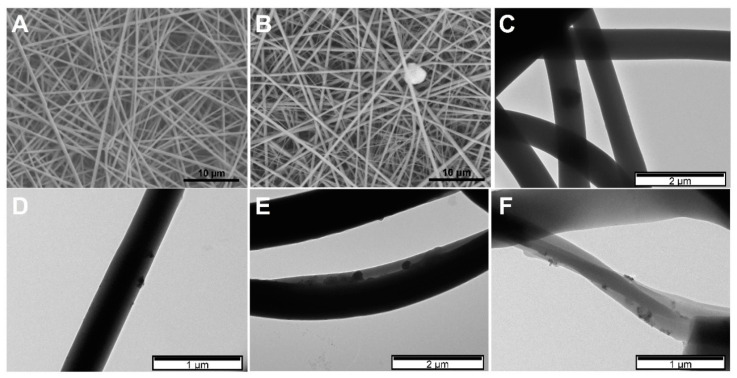
SEM image of bare polyvinylpyrrolidone (PVP) membrane (**A**), and PVP membrane containing PNIPAAm microgels (**B**). Scale bar corresponds to 10 µm. (**C**) TEM image of PVP electrospun fibers containing plain PNIPAAm MG, PNIPAAm MG with bare Fe_3_O_4_ NPs (**D**), PNIPAAm MG with Fe_3_O_4_-OA NPs (**E**), and PNIPAAm MG with Fe_3_O_4_-DMSA NPs (**F**).

**Figure 6 gels-07-00028-f006:**
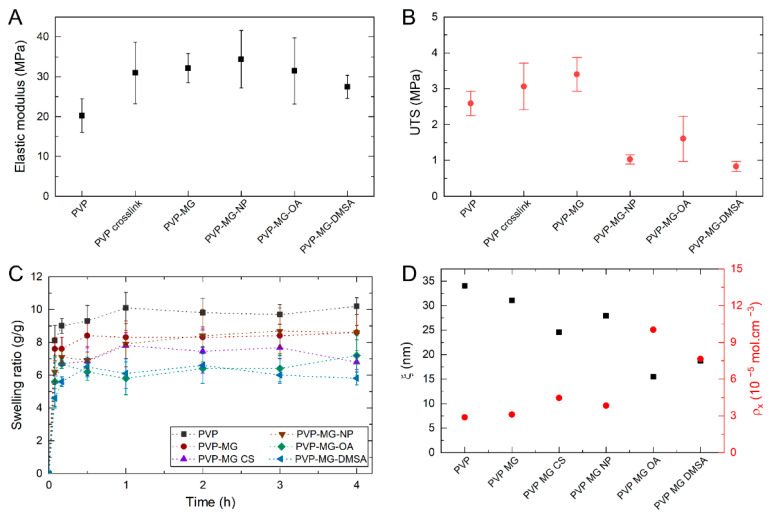
Mechanical parameters of the produced membranes before crosslinking (PVP), after crosslinking (PVP crosslink) and containing 10 wt.% PNIPAAm microgels (PVP-MG): Young’s modulus—*E* (**A**) and ultimate tensile strength—UTS (**B**). (**C**) Swelling behavior, and (**D**) mesh size (ξ) and crosslinking degree (ρ_x_) of the produced PVP membranes without and with PNIPAAm microgels (MG). (PVP-MG CS: PVP membrane containing PNIPAAm microgels with chitosan; PVP-MG-NP: PVP membrane containing PNIPAAm microgels with bare Fe_3_O_4_ NPs; PVP-MG-OA: PVP membrane containing PNIPAAm microgels with Fe_3_O_4_ NPs coated with oleic acid; PVP-MG-DMSA: PVP membrane containing PNIPAAm microgels with Fe_3_O_4_ NPs coated with dimercaptosuccinic acid).

**Figure 7 gels-07-00028-f007:**
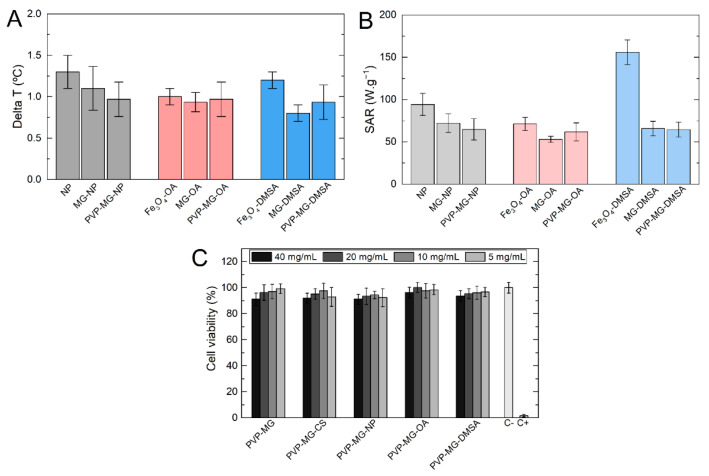
Temperature variation (**A**) and specific absorption ratio (SAR) of PVP membranes with magnetic PNIPAAm microgels incorporated, and the respective microgel and bare Fe_3_O_4_ NPs sample (**B**). Experiments were performed during 10 min of exposure to an alternating magnetic field with an intensity of 24 kA·m^−1^ and 418.5 kHz. (**C**) Vero cell viability (%) after 48 h of indirect exposure to the produced membranes. Initial concentration (C_i_) of 40 mg·mL^−1^ with three factor 2 dilutions (20, 10, and 5 mg·mL^−1^). Data is expressed as average ± standard deviation for five replicas.

**Table 1 gels-07-00028-t001:** $\bar{M_{c}},$ξ and ρ_x_ values for plain and composite PVP membranes.

	PVP	PVP-MG	PVP-MG-CS	PVP-MG-NP	PVP-MG-OA	PVP-MG-DMSA
$\bar{M_{n}}$ (kDa)	44.1	40.9	28.5	33.3	12.7	16.7
ξ (nm)	34.0	31.1	24.6	27.9	15.5	18.8
ρ_x_ $\left(10^{-5}\;\mathrm{mol}\cdot {\mathrm{cm}}^{-1}\right)$	2.9	3.1	4.5	3.8	10.0	7.7

## Data Availability

Data is contained within the article.

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
