# Peer review of "Incorporation of Dual-Stimuli Responsive Microgels in Nanofibrous Membranes for Cancer Treatment by Magnetic Hyperthermia"

_gels, 2021, doi:10.3390/gels7010028_

Round 1

Reviewer 1 Report

This paper needs the improvement of English style, particularly in grammar, before acceptance.

Author Response

The paper was revised, and the English style was improved.

Reviewer 2 Report

The work developed in this manuscript describes the synthesis of dual-stimuli responsive PNIPAAm-NPs microgels and their incorporation in electrospun polymeric fibers.

The methodology is based on those developed in papers previously published by this team, which reduces the novelty of the work presented here.

My main question is what is the interest in having fibres for drug delivery applications. This point could be clarified by the authors in the introduction. Does the intrinsic nature of the objects obtained (fibres) make their application in hyperthermia in humans realistic? Does the fibre form not present toxicity when injected? In this respect, the authors should clarify why they are working on Vero cells.

The cytotoxicity tests carried out only over 48 hours do not, in my opinion, make it possible to rule on the total harmlessness of the membranes. On the other hand, it can be retorted that in cancer therapy, the benefit outweighs the risk, but still. 

Just a remark of form, p5 line 173 it makes write (d/d0)^3 and not (d(d0)^3.

Author Response

The work developed in this manuscript describes the synthesis of dual-stimuli responsive PNIPAAm-NPs microgels and their incorporation in electrospun polymeric fibers:

  1. The methodology is based on those developed in papers previously published by this team, which reduces the novelty of the work presented here.

The methodology is in fact based on the previous work of this team, including the synthesis of PNIPAAm microgels and iron oxide nanoparticles. Although both individual systems were already incorporated in electrospun fibers, the novelty of the present work lays in the development of an electrospun device that holds a combination of both PNIPAAm microgels and magnetic nanoparticles allowing a larger field of applications, in this case, controlled drug delivery and magnetic hyperthermia. To the best of our knowledge, such system was not yet developed.

  1. My main question is what is the interest in having fibers for drug delivery applications. This point could be clarified by the authors in the introduction. Does the intrinsic nature of the objects obtained (fibers) make their application in hyperthermia in humans realistic?

We thank the reviewer for this excellent point. In fact, electrospun fibers have unique characteristics, namely a high surface area to volume ratio, that makes them especially interesting in the drug delivery field. Both the incorporation of PNIPAAm microgels and iron oxide nanoparticles in electrospun nanofibers address the issue of localization of these systems in tumor region in a sufficient concentration to enable effective treatment. Although the temperature variation showed by the produced membranes is not enough to reach a therapeutic temperature, we can observe that they generate approximately the same temperature variation as NPs in solution, that being a good indicator of the efficacy of the composite system, showing the potential of electrospun membranes in hyperthermia applications. In addition to that, the materials used are biocompatible and the cytotoxic assays show no cytotoxicity from the composite membrane, showing it can be used in biomedical applications. Both these results are good indicators that the use of membranes in hyperthermia applications in humans is a realistic approach.

For the sake of clarity, we have included additional information in the introduction section to enhance the advantage of using fibers for controlled drug delivery systems.

  1. Does the fiber form not present toxicity when injected? In this respect, the authors should clarify why they are working on Vero cells. The cytotoxicity tests carried out only over 48 hours do not, in my opinion, make it possible to rule on the total harmlessness of the membranes. On the other hand, it can be retorted that in cancer therapy, the benefit outweighs the risk, but still. 

We apologize for missing some information about the envisage application of our system. In fact, this system is designed to be implanted directly into the tumor site following solid tumor recession. Therefore, no toxicity will be associated with this procedure.

The cytotoxic assays showed in the present work were a preliminary way to show that the produced membranes did not present cytotoxicity and for that we used Vero cells as a standard. We intend to perform more extensive in vitro assays in order to evaluate the complete cytotoxicity and in vitro behavior of the composite membranes.

  1. Just a remark of form, p5 line 173 it makes write (d/d0)^3 and not (d(d0)^3.

We apologize for this minor error; it is now corrected.

Reviewer 3 Report

Gonçalves and coworkers investigated dual-stimuli responsive microgels in nanofibrous membranes for cancer treatment. The text is short, but clear and concise and may be of interest to the readers of Gels. Based on that, I support its publication after some revisions, as stated below:

The resolution of figure 1, 3, and 5 is not enough for publication.

  1. Line 40-43, several studies (doi.org/10.1021/acs.biomac.6b00168; doi.org/10.1016/j.actbio.2019.07.003) should be cited to support such claim.
  2. Line 141-143, the authors claim ‘bare Fe3O4 NPs are located preferably at the microgel surface’. However, in figure 2a, you can’t see any Fe3O4 NPs, and in figure 2b, Fe3O4-OA NPs actually locate at the surfaces of the microgel.
  3. The authors should make the caption of figure 2 more clearer.
  4. It will be better if the authors could add one figure to schematically show the microgels with different NPs. The chemical structures should be included in such figure.
  5. Why the hydrodynamic diameter of PNIPAM microgels decreases upon the LCST? The dehydration will lead to a phase separation, which will result in milky solution.
  6. The zeta potential of the microgels with different NPs should be added.

Author Response

Gonçalves and coworkers investigated dual-stimuli responsive microgels in nanofibrous membranes for cancer treatment. The text is short, but clear and concise and may be of interest to the readers of Gels. Based on that, I support its publication after some revisions, as stated below:

  1. The resolution of figure 1, 3, and 5 is not enough for publication.

As suggested by the reviewer, we submitted higher quality figures together with this revision.

 Line 40-43, several studies (doi.org/10.1021/acs.biomac.6b00168; doi.org/10.1016/j.actbio.2019.07.003) should be cited to support such claim.

We thank the reviewer for the suggestions. Both papers were added to the paper.

  1. Line 141-143, the authors claim ‘bare Fe3O4 NPs are located preferably at the microgel surface’. However, in figure 2a, you can’t see any Fe3O4 NPs, and in figure 2b, Fe3O4-OA NPs actually locate at the surfaces of the microgel.

Figure 2a shows the TEM image of bare PNIPAAm microgels, without the presence of any type of Fe3O4 NPs and figure 2b shows the TEM image of hybrid microgels incorporated with bare Fe3O4 NPs which are located preferably at the microgels’ surface. Figure 2c and 2d correspond to the hybrid PNIPAAm microgels incorporated with Fe3O4-OA NPs and Fe3O4-DMSA NPs, respectively.

  1. The authors should make the caption of figure 2 more clearer.

We thank the reviewer for this suggestion and for the sake of clarity we have improved the caption of Figure 2:

Figure 2. TEM images of bare PNIPAAm microgels (A), hybrid PNIPAAM microgels incorporated with bare Fe3O4 NPs (B), with Fe3O4-OA NPs (C), and with Fe3O4-DMSA NPs (D). (OA: oleic acid; DMSA: dimercaptosuccinic acid).

  1. It will be better if the authors could add one figure to schematically show the microgels with different NPs. The chemical structures should be included in such figure.

We thank the reviewer for the suggestion and we have included a schematic explaining the interaction between nanoparticles and microgels (Figure 3).

  1. Why the hydrodynamic diameter of PNIPAM microgels decreases upon the LCST? The dehydration will lead to a phase separation, which will result in milky solution.

Poly (N-isopropyl acrylamide) (PNIPAAm) has a negative temperature response, with a Lower Critical Solution Temperature (LCST) near human body temperature (30-32 ºC). This means that bellow that temperature the microgels are in their swollen state and collapse when the temperature rises above LCST. The thermoresponsive behavior showed by PNIPAAm emerges from the interactions between its hydrophilic and hydrophobic groups – acrylamide and isopropyl groups, respectively. Below the LCST, the hydrophilic groups of the polymer establish intermolecular hydrogen bonds with water, leading the microgels to be in their swollen and hydrated state. However, as the temperature increases above LCST, these intermolecular bonds between the hydrophilic groups and the solvent weaken, giving rise to intramolecular bonds among hydrophobic segments. The polymer collapses, leading to a phase separation between the microgels and the surrounding medium, thus reducing its volume.

We have added this information also in the introduction section of the paper.

  1. The zeta potential of the microgels with different NPs should be added.

We thank the reviewer for the suggestion. At the moment, due do COVID-19, we are not able to perform zeta potential measurements of the different microgels. However, this is something to have in mind and it will be performed in future works.

Round 2

Reviewer 3 Report

There are still some issues needed to be addressed before publication.

  1. The resolution of figure 1, 3, 4, and 6 is still not enough. Please improve it to the same level of figure 7.
  2. The refs suggeted are not included in the revison.
  3. About the comment 6 from my last report, I was asking why there is no macroscopic phase seperation instead of shrinking the microgels above the LCST. If macroscopic phase seperation happens due to the dehydration, the size from DLS should not become smaller.

Author Response

  1. The resolution of figure 1, 3, 4, and 6 is still not enough. Please improve it to the same level of figure 7.

As suggested by the reviewer, we submitted higher quality figures together with the previous revision. However, our pdf file had low quality. We have improved our pdf file to be in accordance to our images quality.

  1. The refs suggeted are not included in the revison.

We are sorry for this lapse. Now both papers were added to the paper.

  1. About the comment 6 from my last report, I was asking why there is no macroscopic phase seperation instead of shrinking the microgels above the LCST. If macroscopic phase seperation happens due to the dehydration, the size from DLS should not become smaller.

Below LCST, the hydrophilic groups of the polymer establish intermolecular hydrogen bonds with water, leading the microgels to be in their swollen state. At this point, we have larger diameters, but still as stable microgels in an aqueous medium. Above LCST, the dehydration process leads to the break of these bonds with water and the appearance of intramolecular bonds among hydrophobic segments of PNIPAAm itself. This leads to the shrinking process with a separation between the microgels and the solvent. The colloidal particles, although shrink, stay stable in a colloidal solution. Therefore, DLS measurements demonstrate this change in size, thus decreasing the microgels size above LCST. The appearance of a milky solution occurs due to the presence of homogeneously dispersed colloidal particles in the solution.

Further discussion can be found in https://doi.org/10.1016/j.jcis.2010.05.034; DOI: 10.1039/C4SM01222D; doi.org/10.1016/j.jmps.2011.08.008